# YET ANOTHER BUT MORE EFFICIENT BLACK-BOX ADVERSARIAL ATTACK: TILING AND EVOLUTION STRATEGIES

## ABSTRACT

We introduce a new black-box attack achieving state of the art performances. Our approach is based on a new objective function, borrowing ideas from $\ell_\infty$-white box attacks, and particularly designed to fit derivative-free optimization requirements. It only requires to have access to the logits of the classifier without any other information which is a more realistic scenario. Not only we introduce a new objective function, we extend previous works on black box adversarial attacks to a larger spectrum of evolution strategies and other derivative-free optimization methods. We also highlight a new intriguing property that deep neural networks are not robust to single shot tiled attacks. Our models achieve, with a budget limited to $10,000$ queries, results up to $99.2\%$ of success rate against InceptionV3 classifier with $630$ queries to the network on average in the untargeted attacks setting, which is an improvement by $90$ queries of the current state of the art. In the targeted setting, we are able to reach, with a limited budget of $100,000$, $100\%$ of success rate with a budget of $6,662$ queries on average, i.e. we need $800$ queries less than the current state of the art.

## 1 INTRODUCTION

Despite their success, deep learning algorithms have shown vulnerability to adversarial attacks (Biggio et al., 2013; Szegedy et al., 2014), *i.e.* small imperceptible perturbations of the inputs, that lead the networks to misclassify the generated adversarial examples. Since their discovery, adversarial attacks and defenses have become one of the hottest research topics in the machine learning community as serious security issues are raised in many critical fields. They also question our understanding of deep learning behaviors. Although some advances have been made to explain theoretically (Fawzi et al., 2016; Sinha et al., 2017; Cohen et al., 2019; Pinot et al., 2019) and experimentally (Goodfellow et al., 2015; Xie et al., 2018; Meng & Chen, 2017; Samangouei et al., 2018; Araujo et al., 2019) adversarial attacks, the phenomenon remains misunderstood and there is still a gap to come up with principled guarantees on the robustness of neural networks against maliciously crafted attacks. Designing new and stronger attacks helps building better defenses, hence the motivation of our work.

First attacks were generated in a setting where the attacker knows all the information of the network (architecture and parameters). In this *white box* setting, the main idea is to perturb the input in the direction of the gradient of the loss w.r.t. the input (Goodfellow et al., 2015; Kurakin et al., 2016; Carlini & Wagner, 2017; Moosavi-Dezfooli et al., 2016). This case is unrealistic because the attacker has only limited access to the network in practice. For instance, web services that propose commercial recognition systems such as Amazon or Google are backed by pretrained neural networks. A user can *query* this system by sending an image to classify. For such a query, the user only has access to the inference results of the classifier which might be either the label, probabilities or logits. Such a setting is coined in the literature as the *black box* setting. It is more realistic but also more challenging from the attacker's standpoint.

As a consequence, several works proposed black box attacks by just querying the inference results of a given classifier. A natural way consists in exploiting the transferability of an adversarial attack, based on the idea that if an example fools a classifier, it is more likely that it fools another one (Papernot et al., 2016a). In this case, a white box attack is crafted on a fully known classifier. Papernot

et al. (2017) exploited this property to derive practical black box attacks. Another approach within the black box setting consists in estimating the gradient of the loss by querying the classifier (Chen et al., 2017; Ilyas et al., 2018a;b). For these attacks, the PGD attack (Kurakin et al., 2016; Madry et al., 2018a) algorithm is used and the gradient is replaced by its estimation.

In this paper, we propose efficient black box adversarial attacks using stochastic derivative free optimization (DFO) methods with only access to the logits of the classifier. By efficient, we mean that our model requires a limited number of queries while outperforming the state of the art in terms of attack success rate. At the very core of our approach is a new objective function particularly designed to suit classical derivative free optimization. We also highlight a new intriguing property that deep neural networks are not robust to single shot tiled attacks. It leverages results and ideas from $\ell_\infty$-attacks. We also explore a large spectrum of evolution strategies and other derivative-free optimization methods thanks to the Nevergrad framework (Rapin & Teytaud, 2018).

**Outline of the paper.** We present in Section 2 the related work on adversarial attacks. Section 3 presents the core of our approach. We introduce a new generic objective function and discuss two practical instantiations leading to a discrete and a continuous optimization problems. We then give more details on the best performing derivative-free optimization methods, and provide some insights on our models and optimization strategies. Section 4 is dedicated to a thorough experimental analysis, where we show we reach state of the art performances by comparing our models with the most powerful black-box approaches on both targeted and untargeted attacks. We also assess our models against the most efficient so far defense strategy based on adversarial training. We finally conclude our paper in Section 5.

## 2 RELATED WORK

Adversarial attacks have a long standing history in the machine learning community. Early works appeared in the mid 2000's where the authors were concerned about Spam classification (Biggio et al., 2009). Szegedy et al. (2014) revives this research topic by highlighting that deep convolutional networks can be easily fooled. Many adversarial attacks against deep neural networks have been proposed since then. One can distinguish two classes of attacks: white box and black box attacks. In the white box setting, the adversary is supposed to have full knowledge of the network (architecture and parameters), while in the black box one, the adversary only has limited access to the network: she does not know the architecture, and can only query the network and gets labels, logits or probabilities from her queries. An attack is said to have *suceeded* (we also talk about Attack Success Rate), if the input was originally well classified and the generated example is classified to the targeted label.

The white box setting attracted more attention even if it is the more unrealistic between the two. The attacks are crafted by by back-propagating the gradient of the loss function w.r.t. the input. The problem writes as a non-convex optimization procedure that either constraints the perturbation or aims at minimizing its norm. Among the most popular ones, one can cite FGSM (Goodfellow et al., 2015), PGD (Kurakin et al., 2016; Madry et al., 2018a), Deepfool (Moosavi-Dezfooli et al., 2016), JSMA (Papernot et al., 2016b), Carlini&Wagner attack (Carlini & Wagner, 2017) and EAD (Chen et al., 2018).

The black box setting is more realistic, but also more challenging. Two strategies emerged in the literature to craft attacks within this setting: transferability from a substitute network, and gradient estimation algorithms. Transferability has been pointed out by Papernot et al. (2017). It consists in generating a white-box adversarial example on a fully known substitute neural network, i.e. a network trained on the same classification task. This crafted adversarial example can be *transferred* to the targeted unknown network. Leveraging this property, Moosavi-Dezfooli et al. (2017) proposed an algorithm to craft a single adversarial attack that is the same for all examples and all networks. Despite the popularity of these methods, gradient estimation algorithms outperform transferability methods. Chen et al. (2017) proposed a variant of the powerful white-box attack introduced in (Carlini & Wagner, 2017), based on gradient estimation with finite differences. This method achieves good results in practice but requires a high number of queries to the network. To reduce the number of queries, Ilyas et al. (2018a) proposed to rely rather on Natural Evolution Strategies (NES). These derivative-free optimization approaches consist in estimating the parametric distribution of the min-

ima of a given objective function. This amounts for most of NES algorithms to perform a natural gradient descent in the space of distributions (Ollivier et al., 2017). In (Al-Dujaili & O'Reilly, 2019), the authors propose to rather estimate the sign of the gradient instead of estimating the its magnitude suing zeroth-order optimization techniques. They show further how to reduce the search space from exponential to linear. The achieved results were state of the art at the publication date. In Liu et al. (2019), the authors introduced a zeroth-order version of the signSGD algorithm, studied its convergence properties and showed its efficiency in crafting adversarial black-box attacks. The results are promising but fail to beat the state of the art. In Tu et al. (2019), the authors introduce the AutoZOOM framework combining gradient estimation and an auto-encoder trained offline with unlabeled data. The idea is appealing but requires training an auto-encoder with an available dataset, which an additional effort for the attacker. Besides, this may be unrealistic for several use cases. More recently, Moon et al. (2019) proposed a method based on discrete and combinatorial optimization where the perturbations are pushed towards the corners of the $\ell_\infty$ ball. This method is to the best of our knowledge the state of the art in the black box setting in terms of queries budget and success rate. We will focus in our experiments on this method and show how our approaches achieve better results.

Several defense strategies have been proposed to diminish the impact of adversarial attacks on networks accuracies. A basic workaround, introduced in (Goodfellow et al., 2015), is to augment the learning set with adversarial attacks examples. Such an approach is called adversarial training in the literature. It helps recovering some accuracy but fails to fully defend the network, and lacks theoretical guarantees, in particular principled certificates. Defenses based on randomization at inference time were also proposed (Lecuyer et al., 2018; Cohen et al., 2019; Pinot et al., 2019). These methods are grounded theoretically, but the guarantees cannot ensure full protection against adversarial examples. The question of defenses and attacks is still widely open since our understanding of this phenomenon is still in its infancy. We evaluate our approach against adversarial training, the most powerful defense method so far.

# 3 METHODS

## 3.1 GENERAL FRAMEWORK

Let us consider a classification task $\mathcal{X} \mapsto [K]$ where $\mathcal{X} \subseteq \mathbb{R}^d$ is the input space and $[K] = \{1, ..., K\}$ is the corresponding label set. Let $f : \mathbb{R}^d \to \mathbb{R}^K$ be a classifier (a feed forward neural network in our paper) from an input space $\mathcal{X}$ returning the logits of each label in $[K]$ such that the predicted label for a given input is $\arg\max_{i \in [K]} f_i(x)$. The aim of $||.||_\infty$-bounded untargeted adversarial attacks is, for some input $x$ with label $y$, to find a perturbation $\tau$ such that $\arg\max_{i \in [K]} f_i(x) \neq y$. Classically, $||.||_\infty$-bounded untargeted adversarial attacks aims at optimizing the following objective:

$$\max_{\tau : ||\tau||_\infty \leq \epsilon} L(f(x + \tau), y) \tag{1}$$

where $L$ is a loss function (typically the cross entropy) and $y$ the true label. For targeted attacks, the attacker targets a label $y_t$ by maximizing $-L(f(x + \tau), y_t)$. With access to the gradients of the network, gradient descent methods have proved their efficiency (Kurakin et al., 2016; Madry et al., 2018a). So far, the outline of most black box attacks was to estimate the gradient using either finite differences or natural evolution strategies. Here using evolutionary strategies heuristics, we do not want to take care of the gradient estimation problem.

## 3.2 TWO OPTIMIZATION PROBLEMS

In some DFO approaches, the default search space is $\mathbb{R}^d$. In the $\ell_\infty$ bounded adversarial attacks setting, the search space is $B_\infty(\epsilon) = \{\tau : ||\tau||_\infty \leq \epsilon\}$. It requires to adapt the problem in Eq 1. Two variants are proposed in the sequel leading to continuous and discretized versions of the problem.

**The continuous problem.** As in Carlini & Wagner (2017), we use the hyperbolic tangent transformation to restate our problem since $B_\infty(\epsilon) = \epsilon \tanh(\mathbb{R}^d)$. This leads to a continuous search space on which evolutionary strategies apply. Hence our optimization problem writes:

$$\max_{\tau \in \mathbb{R}^d} L(f(x + \epsilon \tanh(\tau)), y). \tag{2}$$

We will call this problem $\mathrm{DFO}_c - \mathrm{optimizer}$ where $\mathrm{optimizer}$ is the used black box derivative free optimization strategy.

**The discretized problem.** Moon et al. (2019) pointed out that PGD attacks (Kurakin et al., 2016; Madry et al., 2018b) are mainly located on the corners of the $\ell_\infty$-ball. They consider optimizing the following

$$\max_{\tau \in \{-\epsilon, +\epsilon\}^d} L(f(x + \tau), y). \tag{3}$$

The author in (Moon et al., 2019) proposed a purely discrete combinatorial optimization to solve this problem (Eq. 3). As in Bello et al. (2017), we here consider how to automatically convert an algorithm designed for continuous optimization to discrete optimization. To make the problem in Eq. 3 compliant with our evolutionary strategies setting, we rewrite our problem by considering a stochastic function $f(x + \epsilon\tau)$ where, for all $i$, $\tau_i \in \{-1, +1\}$ and $\mathbb{P}(\tau_i = 1) = \mathrm{Softmax}(a_i, b_i) = \frac{e^{a_i}}{e^{a_i} + e^{b_i}}$. Hence our problem amounts to find the best parameters $a_i$ and $b_i$ that optimize:

$$\min_{a,b} \mathbb{E}_{\tau \sim \mathbb{P}_{a,b}} (L(f(x + \epsilon\tau), y)) \tag{4}$$

We then rely on evolutionary strategies to find the parameters $a$ and $b$. As the optima are deterministic, the optimal values for $a$ and $b$ are at infinity. Some ES algorithms are well suited to such setting as will be discussed in the sequel. We will call this problem $\mathrm{DFO}_d - \mathrm{optimizer}$ where $\mathrm{optimizer}$ is the used black box derivative free optimization strategy for $a$ and $b$. In this case, one could reduce the problem to one variale $a_i$ with $\mathbb{P}(\tau_i = 1) = \frac{1}{1 + e^{-a_i}}$, but experimentally the results are comparable, so we concentrate on Problem 4.

### 3.3 DERIVATIVE-FREE OPTIMIZATION METHODS

Derivative-free optimization methods are aimed at optimizing an objective function without access to the gradient. There exists a large and wide literature around derivative free optimisation. In this setting, one algorithm aims to minimize some function $f$ on some space $\mathcal{X}$. The only thing that could be done by this algorithm is to query for some points $x$ the value of $f(x)$. As evaluating $f$ can be computationally expensive, the purpose of DFO methods is to get a good approximation of the optima using a moderate number of queries. We tested several evolution strategies (Rechenberg, 1973; Beyer, 2001): the simple $(1 + 1)$-algorithm (Matyas, 1965; Schumer & Steiglitz, 1968), Covariance Matrix Adaptation (CMA (Hansen & Ostermeier, 2003)). For these methods, the underlying algorithm is to iteratively update some distribution $P_\theta$ defined on $\mathcal{X}$. Roughly speaking, the current distribution $\mathbb{P}_\theta$ represents the current belief of the localization of the optimas of the goal function. The parameters are updated using objective function values at different points. It turns out that this family of algorithms, than can be reinterpreted as natural evolution strategies, perform best. The two best performing methods will be detailed in Section 3.3.1; we refer to references above for other tested methods.

#### 3.3.1 OUR BEST PERFORMING METHODS: EVOLUTION STRATEGIES

**The $(1 + 1)$-ES algorithm.** The $(1 + 1)$-evolution strategy with one-fifth rule (Matyas, 1965; Schumer & Steiglitz, 1968) is a simple but effective derivative-free optimization algorithm (in supplementary material, Alg. 1). Compared to random search, this algorithm moves the center of the Gaussian sampling according to the best candidate and adapts its scale by taking into account their frequency. Yao & Liu (1996) proposed the use of Cauchy distributions instead of classical Gaussian sampling. This favors large steps, and improves the results in case of (possibly partial) separability of the problem, i.e. when it is meaningful to perform large steps in some directions and very moderate ones in the other directions.

**CMA-ES algorithm.** The Covariance Matrix Adaptation Evolution Strategy (Hansen & Ostermeier, 2003) combines evolution strategies (Beyer, 2001), Cumulative Step-Size Adaptation (Arnold & Beyer, 2004), and a specific method for adaptating the covariance matrix. An outline is provided in supplementary material, Alg. 2. CMA-ES is an effective and robust algorithm, but it becomes catastrophically slow in high dimension due to the expensive computation of the square root of the matrix. As a workaround, Ros & Hansen (2008) propose to approximate the covariance matrix by a diagonal one. This leads to a computational cost linear in the dimension, rather than the original quadratic one.

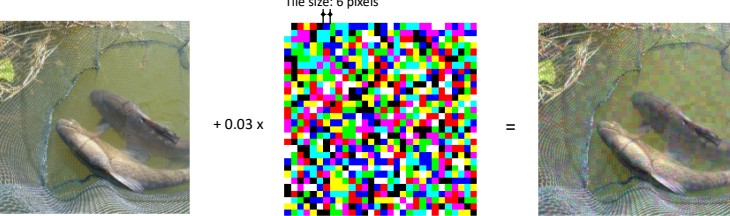

Figure 1: Illustration of the tiling trick: the same noise is applied on small tile squares.

**Link with Natural Evolution Strategy (NES) attacks.** Both (1+1)-ES and CMA-ES can be seen as an instantiation of a natural evolution strategy (see for instance Ollivier et al. (2017); Wierstra et al. (2014)). A natural evolution strategy consists in estimating iteratively the distribution of the optima. For most NES approaches, a fortiori CMA-ES, the iterative estimation consists in a second-order gradient descent (also known as natural gradient) in the space of distributions (e.g. Gaussians). (1+1)-ES can also be seen as a NES, where the covariance matrix is restricted to be proportional to the identity. Note however that from an algorithmic perspective, both CME-ES and (1+1)-ES optimize the quantile of the objective function.

### 3.3.2 HYPOTHESES FOR DFO METHODS IN THE ADVERSARIAL ATTACKS CONTEXT

The state of the art in DFO and intuition suggest the followings. Using $\mathrm{softmax}$ for exploring only points in the corner (Eq. 3) is better for moderate budget, as corners are known to be good adversarial candidates; however, for high precision attacks (with small $\tau$) a smooth continuous precision (Eq 2) is more relevant. With or without $\mathrm{softmax}$, the optimum is at infinity [1], which is in favor of methods having fast step-size adaptation or samplings with heavy-tail distributions. With an optimum at infinity, (Chotard et al., 2012) has shown how fast is the adaptation of the step-size when using cumulative step-size adaptation (as in CMA-ES), as opposed to slower rates for most methods. Cauchy sampling (Yao & Liu, 1996) in the $(1+1)$-ES is known for favoring fast changes; this is consistent with the superiority of Cauchy sampling in our setting compared to Gaussian sampling.

Newuoa, Powell, SQP, Bayesian Optimization, Bayesian optimization are present in Nevergrad but they have an expensive (budget consumption linear is linear w.r.t. the dimension) initial sampling stage which is not possible in our high-dimensional / moderate budget context. The targeted case needs more precision and favors algorithms such as Diagonal CMA-ES which adapt a step-size per coordinate whereas the untargeted case is more in favor of fast random exploration such as the $(1+1)$-ES. Compared to Diagonal-CMA, CMA with full covariance might be too slow; given a number of queries (rather than a time budget) it is however optimal for high precision.

### 3.4 THE TILING TRICK

Ilyas et al. (2018b) suggested to tile the attack to lower the number of queries necessary to fool the network. Concretely, they observe that the gradient coordinates are correlated for close pixels in the images, so they suggested to add the same noise for small square tiles in the image (see Fig. 1). We exploit the same trick since it reduces the dimensionality of the search space, and makes hence evolutionary strategies suited to the problem at hand. Besides breaking the curse of dimensionality, tiling leads surprisingly to a new property that we discovered during our experiments. At a given tiling scale, convolutional neural networks are not robust to random noise. Section 4.2 is devoted to this intriguing property. Interestingly enough, initializing our optimization algorithms with a tiled noise at the appropriate scale drastically speeds up the convergence, leading to a reduced number of queries.

---

[1]i.e. the optima of the ball constrained problem 1, would be close to the boundary or on the boundary of the $\ell_\infty$ ball. In that case, the optimum of the continuous problem 2 will be at $\infty$ or close to it. On the discrete case 4 it is easy to see that the optimum is when $a_i$ or $b_i \to \infty$.

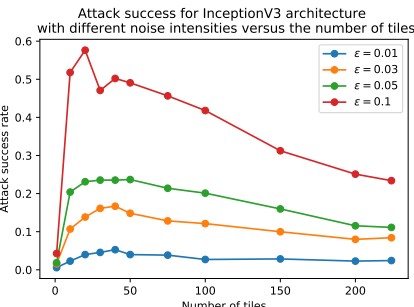 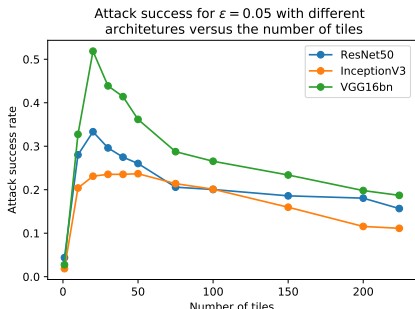

Figure 2: Success rate of a single shot random attacks on ImageNet vs. the number of tiles used to craft the attack. On the left, attacks are plotted against InceptionV3 classifier with different noise intensities ($\epsilon \in \{0.01, 0.03, 0.05, 0.1\}$). On the right, $\epsilon$ is fixed to $0.05$ and the single shot attack is evaluated on InceptionV3, ResNet50 and VGG16bn.

## 4 EXPERIMENTS

### 4.1 GENERAL SETTING AND IMPLEMENTATION DETAILS

We compare our approach to the "bandits" method (Ilyas et al., 2018b) and the parsimonious attack (Moon et al., 2019). The latter (parsimonious attack) is, to the best of our knowledge, the state of the art in the black-box setting from the literature; bandits method is also considered in our benchmark given its ties to our models. We reproduced the results from (Moon et al., 2019) in our setting for fair comparison. As explained in section 3.2, our attacks can be interpreted as $\ell_\infty$ ones. We use the large-scale ImageNet dataset (Deng et al., 2009). As usually done in most frameworks, we quantify our success in terms of attack success rate, median queries and average queries. Here, the number of queries refers to the number of requests to the output logits of a classifier for a given image. For the success rate, we only consider the images that were correctly classified by our model. We use InceptionV3 (Szegedy et al., 2017) , VGG16 (Simonyan & Zisserman, 2014) with batch normalization (VGG16bn) and ResNet50 (He et al., 2016) architectures to measure the performance of our algorithm on the ImageNet dataset. These models reach accuracy close to the the state of the art with around $75 - 80\%$ for the Top-1 accuracy and $95\%$ for the Top-5 accuracy. We use pretrained models from PyTorch (Paszke et al., 2017). All images are normalized to $[0, 1]$. Results on VGG16bn and ResNet50 are deferred in supplementary material E. The images to be attacked are selected at random.

We first show that convolutional networks are not robust to tiled random noise, and more surprisingly that there exists an optimal tile size that is the same for all architectures and noise intensities. Then, we evaluate our methods on both targeted and untargeted objectives. We considered the following losses: the cross entropy $L(f(x), y) = -\log(\mathbb{P}(y|x))$ and a loss inspired from the "Carlini&Wagner" attack: $L(f(x), y) = -\mathbb{P}(y|x) + \max_{y' \neq y} \mathbb{P}(y'|x)$ where $\mathbb{P}(y|x) = [\text{Softmax}(f(x))]_y$, the probability for the classifier to classify the input $x$ to label $y$. The results for the second loss are deferred in supplementary material C. For all our attacks, we use the Nevergrad (Rapin & Teytaud, 2018) implementation of evolution strategies. We did not change the default parameters of the optimization strategies.

### 4.2 CONVOLUTIONAL NEURAL NETWORKS ARE NOT ROBUST TO TILED RANDOM NOISE

In this section, we highlight that neural neural networks are not robust to $\ell_\infty$ tiled random noise. A noise on an image is said to be tiled if the added noise on the image is the same on small squares of pixels (see Figure 2). In practice, we divide our image in equally sized tiles. For each tile, we add to the image a randomly chosen constant noise: $+\epsilon$ with probability $\frac{1}{2}$ and $-\epsilon$ with probability $\frac{1}{2}$, uniformly on the tile. The tile trick has been introduced inIlyas et al. (2018a) for dimensionality reduction. Here we exhibit a new behavior that we discovered during our experiments. As shown in Fig. 1 for reasonable noise intensity ($\epsilon = 0.05$), the success rate of a one shot randomly tiled attack is quite high. This fact is observed on many neural network architectures. We compared the

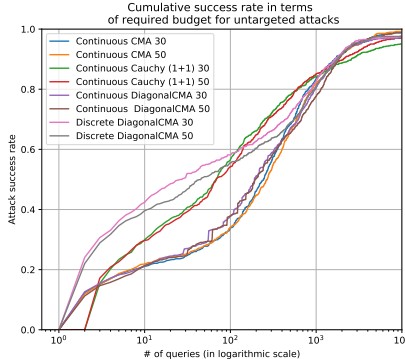 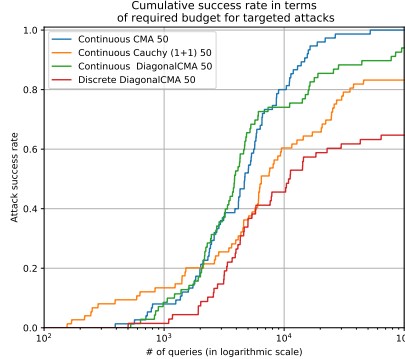

Figure 3: The cumulative success rate in terms the number of queries for the number of queries required for attacks on ImageNet with $\epsilon = 0.05$ in the untargeted (left) and targeted setting (right). The number of queries (x-axis) is plotted with a logarithmic scale.

Table 1: Comparison of our method with the parsimonious and bandits attacks in the untargeted setting on ImageNet on InceptionV3 pretrained network for $\epsilon = 0.05$ and $10,000$ as budget limit.

| Method | # of tiles | Average queries | Median queries | Success rate |
|---|---|---|---|---|
| Parsimonious | - | 702 | 222 | 98.4% |
| Bandits | 30 | 1007 | 269 | 95.3% |
| Bandits | 50 | 995 | 249 | 95.1% |
| $DFO_c - \text{Cauchy}(1+1)\text{-ES}$ | 30 | 466 | 60 | 95.2% |
| $DFO_c - \text{Cauchy}(1+1)\text{-ES}$ | 50 | 510 | 63 | 97.3% |
| $DFO_c - \text{DiagonalCMA}$ | 30 | 533 | 189 | 97.2% |
| $DFO_c - \text{DiagonalCMA}$ | 50 | 623 | 191 | 98.7% |
| $DFO_c - \text{CMA}$ | 30 | 589 | 232 | 98.9% |
| $DFO_c - \text{CMA}$ | 50 | 630 | 259 | **99.2%** |
| $DFO_d - \text{DiagonalCMA}$ | 30 | **424** | **20** | 97.7% |
| $DFO_d - \text{DiagonalCMA}$ | 50 | 485 | 38 | 97.4% |

number of tiles since the images input size are not the same for all architectures ($299 \times 299 \times 3$ for InceptionV3 and $224 \times 224 \times 3$ for VGG16bn and ResNet50). The optimal number of tiles (in the sense of attack success rate) is, surprisingly, independent from the architecture and the noise intensity. We also note that the InceptionV3 architecture is more robust to random tiled noise than VGG16bn and ResNet50 architectures. InceptionV3 blocks are parallel convolutions with different filter sizes that are concatenated. Using different filter sizes may attenuate the effect of the tiled noise since some convolution sizes might be less sensitive. We test this with a single random attack with various numbers of tiles (cf. Figure 1, 2). We plotted additional graphs in supplementary material B.

### 4.3 UNTARGETED ADVERSARIAL ATTACKS

We first evaluate our attacks in the untargeted setting. The aim is to change the predicted label of the classifier. Following (Moon et al., 2019; Ilyas et al., 2018b), we use $10,000$ images that are initially correctly classified and we limit the budget to $10,000$ queries. We experimented with 30 and 50 tiles on the images. Only the best performing methods are reported in Table 1. We compare our results with (Moon et al., 2019) and (Ilyas et al., 2018b) on InceptionV3 (cf. Table 1). We also plotted the cumulative success rate in terms of required budget in Figure 3. We also evaluated our attacks for smaller noise in supplementary material D. We achieve results outperforming or at least equal to the state of the art in all cases. More remarkably, We improve by far the number of necessary queries to fool the classifiers. The tiling trick partially explains why the average and the median number of queries are low. Indeed, the first queries of our evolution strategies is in general close to random

Table 2: Comparison of our method with the parsimonious and bandits attacks in the targeted setting on ImageNet on InceptionV3 pretrained network for $\epsilon = 0.05$ and $100,000$ as budget limit.

| Method | # of tiles | Average queries | Median queries | Success rate |
|---|---|---|---|---|
| Parsimonious | - | 7184 | 5116 | 100% |
| Bandits | 50 | 25341 | 18053 | 92.5% |
| $\mathrm{DFO}_c - \mathrm{Cauchy}(1+1)\text{-ES}$ | 50 | 9789 | 6049 | 83.2% |
| $\mathrm{DFO}_c - \mathrm{DiagonalCMA}$ | 50 | 6768 | **3797** | 94.0% |
| $\mathrm{DFO}_c - \mathrm{CMA}$ | 50 | **6662** | 4692 | **100%** |
| $\mathrm{DFO}_d - \mathrm{DiagonalCMA}$ | 50 | 8957 | 4619 | 64.2% |

search and hence, according to the observation of Figs 1-2, the first steps are more likely to fool the network, which explains why the queries budget remains low. This Discrete strategies reach better median numbers of queries - which is consistent as we directly search on the limits of the $\ell_\infty$-ball; however, given the restricted search space (only corners of the search space are considered), the success rate is lower and on average the number of queries increases due to hard cases.

### 4.4 TARGETED ADVERSARIAL ATTACKS

We also evaluate our methods in the targeted case on ImageNet dataset. We selected $1,000$ images, correctly classified. Since the targeted task is harder than the untargeted case, we set the maximum budget to $100,000$ queries, and $\epsilon = 0.05$. We uniformly chose the target class among the incorrect ones. We evaluated our attacks in comparison with the bandits methods (Ilyas et al., 2018b) and the parsimonious attack (Moon et al., 2019) on InceptionV3 classifier. We also plotted the cumulative success rate in terms of required budget in Figure 3. CMA-ES beats the state of the art on all criteria. DiagonalCMA-ES obtains acceptable results but is less powerful that CMA-ES in this specific case. The classical CMA optimizer is more precise, even if the run time is much longer. Cauchy $(1+1)$-ES and discretized optimization reach good results, but when the task is more complicated they do not reach as good results as the state of the art in black box targeted attacks.

### 4.5 UNTARGETED ATTACKS AGAINST AN ADVERSARIALLY TRAINED NETWORK

In this section, we experiment our attacks against a defended network by adversarial training (Goodfellow et al., 2015). Since adversarial training is computationally expensive, we restricted ourselves to the CIFAR10 dataset (Krizhevsky et al., 2009) for this experiment. Image size is $32 \times 32 \times 3$. We adversarially trained a WideResNet28x10 (Zagoruyko & Komodakis, 2016) with PGD $\ell_\infty$ attacks (Kurakin et al., 2016; Madry et al., 2018a) of norm $8/256$ and 10 steps of size $2/256$. In this setting, we randomly selected $1,000$ images, and limited the budget to $20,000$ queries. We ran PGD $\ell_\infty$ attacks (Kurakin et al., 2016; Madry et al., 2018a) of norm $8/256$ and 20 steps of size $1/256$ against our network, and achieved a success rate up to $36\%$, which is the the state of the art in the white box setting. We also compared our method to the Parsimonious and bandit attacks. Results are reported in Appendix 6. On this task, the parsimonious attack method is slightly better than our best approach.

## 5 CONCLUSION

In this paper, we proposed a new framework for crafting black box adversarial attacks based on derivative free optimization. Because of the high dimensionality and the characteristics of the problem (see Section 3.3.2), not all optimization strategies give satisfying results. However, combined with the tiling trick, evolutionary strategies such as CMA, DiagonalCMA and Cauchy (1+1)-ES beats the current state of the art in both targeted and untargeted settings. In particular, $\mathrm{DFO}_c - \mathrm{CMA}$ improves the state of the art in terms of success rate in almost all settings. We also validated the robustness of our attack against an adversarially trained network. Future work will be devoted to better understanding the intriguing property of the effect that a neural network is not robust to a one shot randomly tiled attack.

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

# A    ALGORITHMS

## A.1    THE (1+1)-ES ALGORITHM

---
**Algorithm 1** The $(1+1)$ Evolution Strategy.

---
**Require:** Function $f : \mathbb{R}^d \to \mathbb{R}$ to minimize
  $m \leftarrow 0, C \leftarrow \boldsymbol{I}_d, \sigma \leftarrow 1$
  **for** $t = 1...n$ **do**
    (Generate candidates)
    Generate $m' \sim m + \sigma X$ where $X$ is sampled from a Cauchy or Gaussian distribution.
    **if** $f(m') \leq f(m)$ **then**
      $m \leftarrow m', \sigma \leftarrow 2\sigma$
    **else**
      $\sigma \leftarrow 2^{-\frac{1}{4}}\sigma$
    **end if**
  **end for**

---

## A.2    CMA-ES ALGORITHM

---
**Algorithm 2** CMA-ES algorithm. The $T$ subscript denotes transposition.

---
**Require:** Function $f : \mathbb{R}^d \to \mathbb{R}$ to minimize, parameters $b$, $c$, $w_1 > \ldots, w_\mu > 0$, $p_c$ and others as
  in e.g. (Hansen & Ostermeier, 2003).
  $m \leftarrow 0, C \leftarrow \boldsymbol{I}_d, \sigma \leftarrow 1$
  **for** $t = 1...n$ **do**
    Generate $x_1, ..., x_\lambda \sim m + \sigma \mathcal{N}(0, C)$.
    Define $x_i'$ the $i^{th}$ best of the $x_i$.
    Update the cumulation for $C$: $p_c \leftarrow$ cumulation of $p_c$, overall direction of progress.
    Update the covariance matrix:

$$C \leftarrow (1-c) \underbrace{C}_{inertia} + \frac{c}{b} \underbrace{(p_c \times p_c^T)}_{\text{overall direction}} + c(1 - \frac{1}{b}) \sum_{i=1}^{\mu} w_i \underbrace{\frac{x_i' - m}{\sigma} \times \frac{(x_i' - m)^T}{\sigma}}_{\text{"covariance" of the } \frac{1}{\sigma}x_i'}$$

    Update mean:

$$m \leftarrow \sum_{i=1}^{\mu} w_i x_{i:\lambda}$$

    Update $\sigma$ by cumulative step-size adaptation (Arnold & Beyer, 2004).
  **end for**

---

## B  ADDITIONAL PLOTS FOR THE TILING TRICK

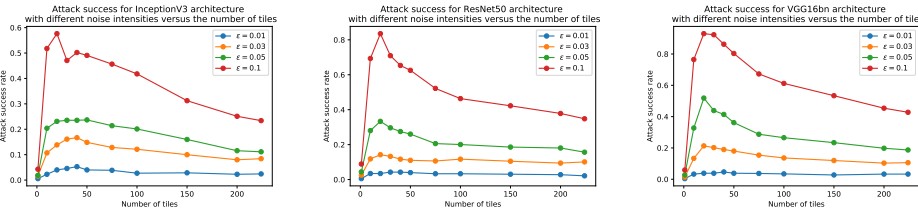

Figure 4: Random attack success rate against InceptionV3 (left), ResNet50 (center), VGG16bn (right) for different noise intensities. We just randomly draw one tiled attack and check if it is successful.

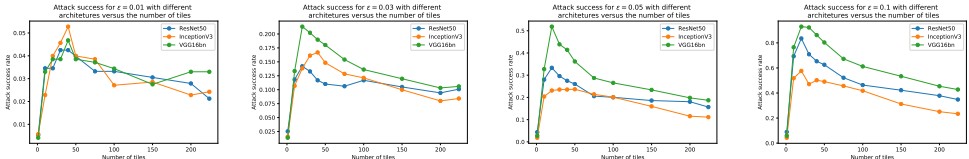

Figure 5: Random attack success rate for different noise intensities $\epsilon \in \{0.01, 0.03, 0.05, 0.1\}$ (from right to left) against different architectures. We just randomly draw one tiled attack and check if it is successful.

## C  RESULTS WITH "CARLINI&WAGNER" LOSS

In this section, we follow the same experimental setup as in Section 4.3, but we built our attacks with the "Carlini&Wagner" loss instead of the cross entropy. We remark the results are comparable and similar.

Table 3: Comparison of our method with"Carlini&Wagner" loss versus the parsimonious and bandits attacks in the untargeted setting on InceptionV3 pretrained network for $\epsilon = 0.05$ and $10,000$ as budget limit.

| Method | # of tiles | Average queries | Median queries | Success rate |
|---|---|---|---|---|
| $\mathrm{DFO}_c - \mathrm{Cauchy}(1+1)\text{-ES}$ | 30 | 353 | 57 | 97.2% |
| $\mathrm{DFO}_c - \mathrm{Cauchy}(1+1)\text{-ES}$ | 50 | **347** | 63 | 98.8% |
| $\mathrm{DFO}_c - \mathrm{DiagonalCMA}$ | 30 | 483 | 167 | 98.8% |
| $\mathrm{DFO}_c - \mathrm{DiagonalCMA}$ | 50 | 528 | 181 | 99.2% |
| $\mathrm{DFO}_c - \mathrm{CMA}$ | 30 | 475 | 225 | 99.2% |
| $\mathrm{DFO}_c - \mathrm{CMA}$ | 50 | 491 | 246 | **99.4%** |
| $\mathrm{DFO}_d - \mathrm{DiagonalCMA}$ | 30 | 482 | **27** | 98.0% |
| $\mathrm{DFO}_d - \mathrm{DiagonalCMA}$ | 50 | 510 | 37 | 98.0% |

# D    UNTARGETED ATTACKS WITH SMALLER NOISE INTENSITIES

We evaluated our method on smaller noise intensities ($\epsilon \in \{0.01, 0.03, 0.05\}$) in the untargeted setting on ImageNet dataset. In this framework, we also picked up randomly $10,000$ images and limited our budget to $10,000$ queries. We compared to the bandits method (Ilyas et al., 2018b) and to the parsimonious attack (Moon et al., 2019) on InceptionV3 network. We limited our experiments to a number of tiles of 50. We report our results in Table 4. We remark our attacks reach state of the art for $\epsilon = 0.03$ and $\epsilon = 0.05$ both in terms of success rate and queries budget. For $\epsilon = 0.01$, we reach results comparable to the state of the art.

Table 4: Results of our method compared to the parsimonious and bandit attacks in the untargeted setting on InceptionV3 pretrained network for different values of noise intensities $\epsilon \in \{0.01, 0.03, 0.05\}$ and a maximum of $10,000$ queries.

| $\epsilon$ | Method | # of tiles | Avg. queries | Med. queries | Success rate |
|---|---|---|---|---|---|
| | Parsimonious | - | 722 | 237 | 98.5% |
| | Bandits | 50 | 995 | 249 | 95.1% |
| 0.05 | $\mathrm{DFO}_c - \mathrm{Cauchy}(1+1)\text{-ES}$ | 50 | 510 | 63 | 97.3% |
| | $\mathrm{DFO}_c - \mathrm{DiagonalCMA}$ | 50 | 623 | 191 | 98.7% |
| | $\mathrm{DFO}_c - \mathrm{CMA}$ | 50 | 630 | 259 | **99.2%** |
| | $\mathrm{DFO}_d - \mathrm{DiagonalCMA}$ | 50 | **485** | **38** | 97.4% |
| | Parsimonious | - | 1104 | 392 | 95.7% |
| | Bandits | 50 | 1376 | 466 | 92.7% |
| 0.03 | $\mathrm{DFO}_c - \mathrm{Cauchy}(1+1)\text{-ES}$ | 50 | 846 | **203** | 93,2% |
| | $\mathrm{DFO}_c - \mathrm{DiagonalCMA}$ | 50 | 971 | 429 | 96,5% |
| | $\mathrm{DFO}_c - \mathrm{CMA}$ | 50 | 911 | 404 | **96.7%** |
| | $\mathrm{DFO}_d - \mathrm{DiagonalCMA}$ | 50 | **799** | 293 | 94,1% |
| | Parsimonious | - | 2104 | 1174 | 80.3% |
| | Bandits | 50 | 2018 | 992 | 72.9% |
| 0.01 | $\mathrm{DFO}_c - \mathrm{Cauchy}(1+1)\text{-ES}$ | 50 | 1668 | **751** | 72,1% |
| | $\mathrm{DFO}_c - \mathrm{DiagonalCMA}$ | 50 | 1958 | 1175 | 79.2% |
| | $\mathrm{DFO}_c - \mathrm{CMA}$ | 50 | 1921 | 1107 | **80.4%** |
| | $\mathrm{DFO}_d - \mathrm{DiagonalCMA}$ | 50 | **1188** | 849 | 71,3% |

# E    UNTARGETED ATTACKS AGAINST OTHER ARCHITECTURES

We also evaluated our method on different neural networks architectures. For each network we randomly selected $10,000$ images that were correctly classified. We limit our budget to $10,000$ queries and set the number of tiles to 50. We achieve a success attack rate up to $100\%$ on every classifier with a budget as low as 8 median queries for the VGG16bn for instance (see Table 5). One should notice that the performances are lower on InceptionV3 as it is also reported for the bandit methods in (Ilyas et al., 2018b). This possibly due to the fact that the tiling trick is less relevant on the Inception network than on the other networks (see Fig. 2).

Table 5: Comparison of our method on the ImageNet dataset with InceptionV3 (I), ResNet50 (R) and VGG16bn (V) for $\epsilon = 0.05$ and $10,000$ as budget limit.

| Method | Tile size | Avg queries | | | Med. queries | | | Succ. Rate | | |
|---|---|---|---|---|---|---|---|---|---|---|
| | | I | R | V | I | R | V | I | R | V |
| $\mathrm{DFO}_c - \mathrm{Cauchy}(1+1)\text{-ES}$ | 30 | 466 | **163** | 86 | 60 | **19** | 8 | 95.2% | 99.6% | **100%** |
| $\mathrm{DFO}_c - \mathrm{Cauchy}(1+1)\text{-ES}$ | 50 | 510 | 218 | **67** | 63 | 32 | 4 | 97.3% | 99.6% | 99.7% |
| $\mathrm{DFO}_c - \mathrm{DiagonalCMA}$ | 30 | 533 | 263 | 174 | 189 | 95 | 55 | 97.2% | 99.0% | 99.9% |
| $\mathrm{DFO}_c - \mathrm{DiagonalCMA}$ | 50 | 623 | 373 | 227 | 191 | 121 | 71 | 98.7% | 99.9% | **100%** |
| $\mathrm{DFO}_c - \mathrm{CMA}$ | 30 | 588 | 256 | 176 | 232 | 138 | 72 | 98.9% | 99.9% | 99.9% |
| $\mathrm{DFO}_c - \mathrm{CMA}$ | 50 | 630 | 270 | 219 | 259 | 143 | 107 | **99.2%** | **100%** | 99.9% |
| $\mathrm{DFO}_d - \mathrm{DiagonalCMA}$ | 50 | 485 | 617 | 345 | 38 | 62 | 6 | 97.4% | 99.2% | 99.6% |
| $\mathrm{DFO}_d - \mathrm{DiagonalCMA}$ | 30 | **424** | 417 | 211 | **20** | 20 | **2** | 97.7% | 98.8% | 99.5% |

# F    TABLE FOR ATTACKS AGAINST ADVERSARIALLY TRANINED NETWORK

Table 6: Adversarial attacks against an adversarially trained WideResnet28x10 network on CIFAR10 dataset for $\epsilon = 0.03125$ and $20,000$ as budget limit.

| Method | # of tiles | Average queries | Median queries | Success rate |
|---|---|---|---|---|
| PGD (not black-box) | - | 20 | 20 | 36% |
| Parsimonious | - | 1130 | 450 | **42%** |
| Bandits | 10 | 1429 | 530 | 29.1% |
| Bandits | 20 | 1802 | 798 | 33.8% |
| Bandits | 32 | 1993 | 812 | 34.8% |
| $DFO_c - Cauchy(1+1)$-ES | 10 | 429 | **60** | 29.5% |
| $DFO_c - Cauchy(1+1)$-ES | 20 | 902 | 93 | 30.5% |
| $DFO_c - Cauchy(1+1)$-ES | 32 | 1865 | 764 | 31.7% |
| $DFO_c - DiagonalCMA$ | 10 | 395 | 85 | 30.5% |
| $DFO_c - DiagonalCMA$ | 20 | 624 | 151 | 31.3% |
| $DFO_c - DiagonalCMA$ | 32 | 1379 | 860 | 34.7% |
| $DFO_c - CMA$ | 10 | **363** | 156 | 30.4% |
| $DFO_c - CMA$ | 20 | 1676 | 740 | 40.2% |
| $DFO_c - CMA$ | 32 | 2311 | 1191 | 40.2% |

## G    FAILING METHODS

In this section, we compare our attacks to other optimization strategies. We run our experiments in the same setup as in Section 4.3. Results are reported in Table 7. DE and Normal (1+1)-ES performs poorly, probably because these optimization strategies converge slower when the optima are at "infinity". We reformulate this sentence accordingly in the updated version of the paper. Finally, as the initialization of Powell is linear with the dimension and with less variance, it performs poorer than simple random search. Newuoa, SQP and Cobyla algorithms have also been tried on a smaller number images (we did not report the results), but their initialization is also linear in the dimension, so they reach very poor results too.

Table 7: Comparison with other DFO optimization strategies in the untargeted setting on ImageNet dataset InceptionV3 pretrained network for $\epsilon = 0.05$ and $10,000$ as budget limit.

| Method | # of tiles | Average queries | Median queries | Success rate |
|---|---|---|---|---|
| $DFO_c - Cauchy(1+1)$-ES | 30 | 466 | 60 | 95.2% |
| $DFO_c - Cauchy(1+1)$-ES | 50 | 510 | 63 | 97.3% |
| $DFO_c - DiagonalCMA$ | 30 | 533 | 189 | 97.2% |
| $DFO_c - DiagonalCMA$ | 50 | 623 | 191 | 98.7% |
| $DFO_c - CMA$ | 30 | 589 | 232 | 98.9% |
| $DFO_c - CMA$ | 50 | 630 | 259 | 99.2% |
| $DFO_c - DE$ | 30 | 756 | 159 | 78.8% |
| $DFO_c - DE$ | 50 | 699 | 149 | 76.0% |
| $DFO_c - Normal(1+1)$-ES | 30 | 581 | 45 | 87.6% |
| $DFO_c - Normal(1+1)$-ES | 50 | 661 | 66 | 92.8% |
| $DFO_c - RandomSearch$ | 30 | 568 | 6 | 37.9% |
| $DFO_c - RandomSearch$ | 50 | 527 | 5 | 38.2% |
| $DFO_c - Powell$ | 30 | 4889 | 5332 | 14.4% |
| $DFO_c - Powell$ | 50 | 4578 | 4076 | 7.3% |

