# OpenReview forum: "Yet another but more efficient black-box adversarial attack: tiling and evolution strategies"
_ICLR.cc/2020/Conference — Reject_

### Official Review · AnonReviewer3 · 2019-10-21
**Official Blind Review #3**

**Rating:** 3

**Review:**

This paper proposed a new query efficient black-box attack algorithm using better evolution strategies. The authors also add tiling trick to make the attack even more efficient. The experimental results show that the proposed method achieves state-of-the-art attack efficiency in black-box setting.

The paper indeed presented slightly better results than the current state-of-the-art black-box attacks. It is clearly written and easy to follow, however, the paper itself does not bring much insightful information.

The major components of the proposed method are two things: using better evolution strategies and using tiling trick. The tiling trick is not something new, it is introduced in (Ilyas et al., 2018) and also discussed in (Moon et al., 2019). The authors further empirically studied the best choice of tiling size. I appreciated that, but will not count it as a major contribution. In terms of better evolution strategies, the authors show that (1+1) and CMA-EA can achieve better attack result but it lacks intuition/explanations why these helps, what is the difference. It would be best if the authors could provide some theories to show the advantages of the proposed method, if not, at least the authors should give more intuition/explanation/demonstrative experiments to show the advantages.

Detailed comments:
- In section 3.2, is the form of the discretized problem a standard way to transform from continuous to discrete one? What is the intuition of using a and b? Have you considered using only one variable to do it?
- In section 3.3.2 what do you mean by “with or without softmax, the optimum is at infinity”? I hope the authors could further explain it.
- In eq (2), do you mean  max_{\tau} L(f(x + \epsilon tanh(\tau)), y) ?
- In section 3.3.1, the authors said (1+1)-ES and CMA-ES can be seen as an instantiation of NES. Can the authors further elaborate on this?
- Can the authors provide algorithm for DiagonalCMA?
- It is better to put the evolution strategy algorithms in the main paper and discuss it.
- Can the authors also comment/compare the results with the following relevant paper?
Li, Yandong, et al. "NATTACK: Learning the Distributions of Adversarial Examples for an Improved Black-Box Attack on Deep Neural Networks." ICML 2019.
Chen, Jinghui, Jinfeng Yi, and Quanquan Gu. "A Frank-Wolfe Framework for Efficient and Effective Adversarial Attacks." arXiv preprint arXiv:1811.10828 (2018).

-  In Table 1, why for Parsimonious and Bandit methods, # of tiles parts are missing? I think both of the baselines use tilting trick? And they should also run using the optimal tiling size? The result seems directly copied from the Parsimonious paper? It makes more sense to rerun it in your setting and environment cause the sampled data points may not be the same. Since CMA costs significantly more time, it makes a fair comparison to also report the attack time needed for each method.

- In Table 3, why did not compare with Bandit and Parsimonious attacks?

======================
after the rebuttal

I thank the authors for their response but I still feel that there is a lot more to improve for this paper in terms of intuition and experiments. Therefore I decided to keep my score unchanged.


**Experience Assessment:**

I have published one or two papers in this area.

**Review Assessment: Checking Correctness Of Derivations And Theory:**

I carefully checked the derivations and theory.

**Review Assessment: Checking Correctness Of Experiments:**

I carefully checked the experiments.

**Review Assessment: Thoroughness In Paper Reading:**

I read the paper thoroughly.

---

> ### Author Response · Authors · 2019-11-11
> **Answer to reviewer 3**
>
> We thank reviewer 3 for its comments.
>
> CMA uses a second order approximation of the shape of level sets. This is computationally expensive, but leads to an optimal use of a restricted budget.
> Diagonal CMA is a computationally faster version, thanks to a diagonal covariance (the reader might think of a diagonal Hessian matrix).
> The (1+1) ES is even simpler; the covariance is proportional to the identity (corresponding to a Hessian matrix with all eigenvalues equal). It is therefore relevant for very low budget as it does not have to learn any matrix; on the other hand it is weaker for greater budget as the sampling does not match the shape of level sets.
>
> - In section 3.2, is the form of the discretized problem a standard way to transform from continuous to discrete one? What is the intuition of using a and b? Have you considered using only one variable to do it?
>
> We designed this formulation ourselves, but it won’t be surprising that it has already been used elsewhere. Using this formulation the solution to (3) is already in the corners of the Linf ball which is intuitively more likely to fool the network,  We tried the two implementations (with one or two variables) and the results are very similar.
>
> - In section 3.3.2 what do you mean by “with or without softmax, the optimum is at infinity”? I hope the authors could further explain it.
>
> Sorry for this unclear statement. The optima of the ball constrained problem (1), would be close to the boundary or on the boundary of the Linf ball.  In that case, the optimum of the continuous problem (2) will be at infty or “close” to it. On the discrete case (3) it is easy to see that the optimum is when a_i or b_i -> infty. We reformulate this sentence accordingly in the updated version of the paper.
>
> - In eq (2), do you mean  max_{\tau} L(f(x + \epsilon tanh(\tau)), y) ?
>
> Thank you for having spotted this typo.
>
> - In section 3.3.1, the authors said (1+1)-ES and CMA-ES can be seen as an instantiation of NES. Can the authors further elaborate on this?
>
> NES strategies are optimizations strategies based on Natural gradient [Ollivier et al., 2017, Wierstra et al., 2008]. It consists in iteratively updating a search distributions (the distribution of the optima). CMA-ES consists in updated the mean and the covariance of the distributions. (1+1)-ES updates the mean and add constraints on the covariance matrix is isotropic. The underlying optimisation is not the function, but its quantiles regarding the distributions.
>
> [Wierstra et al., 2008] https://arxiv.org/pdf/1106.4487.pdf
> [Ollivier et al., 2017] https://arxiv.org/pdf/1106.3708.pdf
>
> - Can the authors provide algorithm for DiagonalCMA?
>
> The DiagonalCMA version is when the updates are only on diagonal coefficients of the covariance matrix, hence making a faster computation. We can provide an algorithm; it is a simple modification of CMA discussed in [Ross & Hansen, 2008] https://hal.inria.fr/inria-00270901/document. We will make it clearer in the paper.
>
> - It is better to put the evolution strategy algorithms in the main paper and discuss it.
>
> Yes we can do so.
>
> - Can the authors also comment/compare the results with the following relevant paper?
> Li et al. "NATTACK: Learning the Distributions of Adversarial Examples for an Improved Black-Box Attack on Deep Neural Networks."
> Chen et al. "A Frank-Wolfe Framework for Efficient and Effective Adversarial Attacks."
>
> For both papers, the reported results are not competitive w.r.t the parsimonious attack. However, we are eager to include them in our benchmark if the reviewer wants so.
>
> -  In Table 1, why for Parsimonious and Bandit methods, # of tiles parts are missing? I think both of the baselines use tilting trick? And they should also run using the optimal tiling size? The result seems directly copied from the Parsimonious paper? It makes more sense to rerun it in your setting and environment cause the sampled data points may not be the same. Since CMA costs significantly more time, it makes a fair comparison to also report the attack time needed for each method.
>
>
> We reported the results from the paper parsimonious, but we did not rerun the experiments in our proper setting, because they use the same architecture. However as suggested by the reviewer, we will re-run the experiments in our setting.
>
> Parsimonious attack divide progressively the image in tiles but do not use a proper tile size. Bandits, itself uses a tile size. In the updated version we make it clearer.
>
> CMA-ES takes indeed quite a lot of time, we will update the paper with the reported runtime - the diagonal one is much faster (but needs more evaluations).
>
> - In Table 3, why did not compare with Bandit and Parsimonious attacks?
>
> In [Moon et al.,2019], they compare on a different architecture which is WideResNet 32x10, as we did not run the experiments in their setting, it is unclear that the results are the same.But we will give the corresponding results in the updated version of our pape.

---

### Official Review · AnonReviewer1 · 2019-10-23
**Official Blind Review #1**

**Rating:** 3

**Review:**

This paper proposes a black box adversarial attacks to deep neural networks. The proposed approaches consist of tiling technique proposed by Ilyas et al (2018) and derivative free approaches. The proposed approaches have been applied to targeted and untargeted adversarial attacks against modern neural network architectures such as VGG16, ResNet50, and InceptionV3 trained on ImageNet and CIFAR10 datasets. Experimental results show higher attack success rate with a smaller number of queries.

The experimental results look quite promising, i.e., revealing the vulnerability of the deep neural network against black-box adversarial attacks. A possible weakness in the experimental design is that the authors haven't apply any defense methodology to the classification models to be attacked. Yet the results are promising.

From the viewpoint of technical soundness, the approach is a simple combination of the existing approaches. The tiling technique is used in Ilyas et al (2018) combined with a bandit approach. The current paper simply replaces the bandit with evolution strategies. The introduction of the evolution strategies is motivated by their good performance as a zeroth order optimization algorithm.

A small novelty appears in a way to handle a bounded search space. The authors claim that many DFO algorithms are designed for unbounded real search space and need some constraint handling. The authors proposed two ways of transforming the bounded search space to the unbounded real search space. However, there must be existing approaches for this type fo constraint (rectangle constraint) in DFO settings. I can not list such approaches here as there are huge number of papers addressing the constraint of this type. There is not enough discussion in the paper why these two proposed approaches are promising. Formulation (2) makes the problem ill-posed and technically the optimal point may not exist. Formulation (3) with softmax representation makes the optimization problem noisy, hence it may annoy the optimizer. Nonetheless, I believe the combination of these constraint handling technique and evolutionary approaches are not new.

Some minor comments / questions below:

P5: How are the original images to be attacked selected for Fig 2?

P6:  "we highlight that neural neural networks are not robust to l∞ tiled random noise. " Isn't it the contribution of (Ilyas et al., 2018b)?

P7: What are the number of queries in Figure 3 and Table 1? Are they the number of queries spent until these algorithms found an adversarial example which is categorized to a wrong class for the first time?

**Experience Assessment:**

I have published one or two papers in this area.

**Review Assessment: Checking Correctness Of Derivations And Theory:**

I assessed the sensibility of the derivations and theory.

**Review Assessment: Checking Correctness Of Experiments:**

I assessed the sensibility of the experiments.

**Review Assessment: Thoroughness In Paper Reading:**

I read the paper at least twice and used my best judgement in assessing the paper.

---

> ### Author Response · Authors · 2019-11-11
> **Answer to reviewer 1**
>
> We thank reviewer 1 for its comment
>
> We applied Adversarial training on CIFAR10 dataset, which is up to our knowledge the most efficient defense method so far.
>
> We agree that many box-constraint handling methods exist. However here the point is not the handling of such constraints: we have that  box constraints and optimal points are close to the frontiers. Based on an extensive set of experiments, using Nevergrad, CMA-ES and (1+1)-ES reveal themselves to be more competitive on this type of problems.
>
> This type of fomulation is not new. In machine learning, this was used e.g. in Zoph et al for instance. We agree that our paper combines existing approaches, even though, up to our knowledge, these types of evolutionary strategies have never been used so far in this context.
> There is a typo in Formulation (2), it is : max_{\tau} L(f(x + \epsilon tanh(\tau)), y).
>
>
> Answers to questions:
>
> P5: How are the original images to be attacked selected for Fig 2?
>
> The images are selected at random in ImageNet dataset
>
> P6:  "we highlight that neural neural networks are not robust to l∞ tiled random noise. " Isn't it the contribution of (Ilyas et al., 2018b)?
>
> Ilyas et al. introduced the tiling trick based on the observation that the gradient does not vary that much for two close points. Here we exhibit that convolutional neural nets are not robust to random tiled noise. This property helps in speeding up a subfamily of evolutionary algorithms based on pure random search in the first steps. This explains the good results obtained by (1+1)-ES and CMA-ES.
>
> P7: What are the number of queries in Figure 3 and Table 1? Are they the number of queries spent until these algorithms found an adversarial example which is categorized to a wrong class for the first time?
>
> In Figure 3, the number of queries are the number of queries spent until our algorithms find an adversarial example. In Table 3, we reported the mean and the median of these numbers of queries. We make it clearer in the updated version of the paper.

---

### Official Review · AnonReviewer2 · 2019-10-24
**Official Blind Review #2**

**Rating:** 3

**Review:**

This paper proposed a DFO framework to generate black-box adversarial examples. By comparing with Parsimonious and Bandits, the proposed approach achieves lower query complexity and higher attack success rate (ASR).

I have two main concerns about the current version:

1)  Some important baselines might be missing. In addition to (Ilyas et al., 2018b) and (Moon et al., 2019), the methods built on zeroth-order optimization (namely, gradient estimation via function differences) were not compared. Examples include
[1] There are No Bit Parts for Sign Bits in Black-Box Attacks
[2] AutoZOOM: Autoencoder-based Zeroth Order Optimization Method for Attacking Black-box Neural Networks
[3] SIGNSGD VIA ZEROTH-ORDER ORACLE

2) In addition to attack success rate and query complexity, it might be useful to compare different attacks in terms of $\ell_p$ distortion, where $p \neq \infty$. This could provide a clearer picture on whether or not the query efficiency and the attack performance are at the cost of increasing the $\ell_1$ and $\ell_2$ distortion significantly.


########### Post-feedback ##############
Thanks for the response and the additional experiments to address my first question.  However, I am not satisfied with the response "But clearly our methods aim to reach the boundary of linf ball, so the distortion might be large" to the second question.

I am Okay with the design of $\ell_\infty$ attack. However, if the reduction in query complexity is at a large cost of perturbation power, e.g., measured by $\ell_2$ norm, then it is better to demonstrate this tradeoff. Furthermore, if the $\ell_2$ norm is constrained, will the proposed $\ell_\infty$ attack outperform the others? This is also not clear to me.

Thus, I decide to keep my score.

**Experience Assessment:**

I have published in this field for several years.

**Review Assessment: Checking Correctness Of Derivations And Theory:**

N/A

**Review Assessment: Checking Correctness Of Experiments:**

I assessed the sensibility of the experiments.

**Review Assessment: Thoroughness In Paper Reading:**

I read the paper at least twice and used my best judgement in assessing the paper.

---

> ### Author Response · Authors · 2019-11-11
> **Answer to reviewer 2**
>
> We thank reviewer 2 for its comments on the paper.
>
> 1. We thank reviewer 2 for pointing out these articles. We decided to compare to what is, up to our knowledge, the state of the art in black-box attacks (at least in published papers in Neurips 2019, 2018, ICML 2019, etc.), which is the Parsimonious attack [Moon et al., 2019]. Bandits is often taken as reference for black-box attacks [Ilyas et al., 2018], so we took it as a reference. We read the papers you provided to us. It comes out that [1] would be a good baseline to compare with too. Note that this paper is  also submitted to ICLR (https://openreview.net/forum?id=SygW0TEFwH&noteId=SJx_zBx6tH) and we were not aware of the existence of this paper, so thank you. The results reported in [3] are not competitive to those obtained by Parsimonious attacks. The attack designed in [2] is an L2 one.  It requires the training of an autoencoder, which is not fair for comparison with black-box attacks our algorithms belong to.
>
>
> 2. The method we propose is for a Linf bounded problem, it is not usual to compare with other distortions. But clearly our methods aim to reach the boundary of linf ball, so the distortion might be large. That's why we also compare to Linf attacks too.

---

> ### Author Response · Authors · 2019-11-13
> **Comment about "There are No Bit Parts for Sign Bits in Black-Box Attacks"**
>
> Compared to SignHunter, for epsilon=0.05, max 10000 queries, and ImageNet Inceptionv3, SignHunter reaches 2% failure rate with average of 578.6 queries, whereas we reach with continuous CMA and tile size of 30, 1.1% failure rate with average of 589 queries. We seem to have slightly better results than they have.

---

### Author Response · Authors · 2019-11-15
**Update of the paper**

We updated a version of the paper according to the remarks of all the three reviewers.

---

### Decision · Program_Chairs · 2019-12-19

**Decision:**

Reject

**Comment:**

This paper proposes a new black-box adversarial attack based on tiling and evolution strategies. While the experimental results look promising, the main concern of the reviewers is the novelty of the proposed algorithm, and many things need to be improved in terms of clarity and experiments. The paper does not gather sufficient support from the reviewers even after author response. I encourage the authors to improve this paper and resubmit to future conference.